# A Hidden Fingerprint Device on an Opaque Display Panel

**Jhe-Syuan Lin and Wen-Shing Sun \***

Department of Optics and Photonics, National Central University, Taoyuan 32001, Taiwan; 101286003@cc.ncu.edu.tw

**\*** Correspondence: wssun@dop.ncu.edu.tw; Tel.: +886-3-4227151 (ext. 65259)

**Abstract:** In recent years, fingerprint recognition has become more and more widely used in mobile phones. A fingerprint recognition device hidden under an opaque display panel designed based on a waveguide and frustrated total internal reflection (FTIR) is proposed and demonstrated herein. In order to meet the demand for a high screen ratio for mobile phone displays, we use a symmetrical zoom-in and zoom-out coupler design. With this comprehensive coupler and waveguide design, not only can fingerprint recognition be achieved using an opaque display panel, but it also meets the appearance requirements for a mobile phone with a high screen ratio.

**Keywords:** fingerprint recognition; optical waveguide; opaque display panel

## 1. Introduction

There are several biometric recognition systems that have been developed for applications in mobile phones, such as iris, facial and fingerprint recognition [1,2]. Among them, fingerprint identification is very important method because of its permanence, uniqueness and reproducibility. It can be achieved by ultrasonic [3], capacitive and optical methods [4]. Optical fingerprint recognition (OFR) is the most widely used in access control systems because of its high image resolution and anti-spoofing capabilities [4–7]. However, OFR cannot be used on mobile phones with opaque type displays (such as liquid crystal displays, LCDs) due to the following three main reasons. First, the device is hard to integrate into a mobile phone due to its bulkiness [1]. For example, optical coherence tomography (OCT) OFR technology [8] can obtain good quality 3D fingerprint images and information, but complex and large optical architectures make it impossible to integrate into mobile phones. Secondly, the backlight module of the LCD panel is opaque, so OFR has difficulty penetrating the LCD panel directly. So far, optical fingerprints can only be used on mobile phones equipped with OLED displays. A wide angle, 3-plastic aspherical lens (total thickness <4 mm) is placed under the OLED display directly, and then the fingerprint on the screen is taken through this camera. Since the fingerprint image quality is affected by screen obscuration, the fingerprint image must be enhanced by special image processing. Thus, an LCD display phone can only use capacitive fingerprint unlocking solutions for now.

The last issue is the appearance requirements for a mobile phone with a high screen ratio. Although there are some optical methods to overcome opaque display panels, they cannot meet the requirements of high screen ratios. Due to the reasons above, capacitive fingerprint methods are only available on opaque display phones [9,10].

In this paper, we have developed a method to overcome these challenges. The integration of a waveguide into the cover glass of the mobile phone can solve the problem of it not directly being able to penetrate the opaque display, and the concept of FTIR is applied to the cover glass to enhance the contrast of the fingerprint image. The concept of combining cover glass and FTIR has been applied to the design of touch panels [11]. Cameras are used to capture light scattered by the ridge of the finger,

and image processing techniques are used to detect the position of finger pressing. However, we use the unscattered light in the cover glass to form the image of finger ridges and valleys. Furthermore, using the concept of optical projection magnification, the light beam diameter can reach the minimum diameter when entering and exiting the cover glass. This optical design concept can be applied used to meet the need for a high screen ratio.

## 2. Theory

A prototype of the proposed OFR that can be integrated into a display is presented in Figure 1a. Light from the collimated light source system [12] is reflected into the cover glass by the reflective surface above it, as shown in Figure 1b. The light area is projected onto the reflecting surface and changed from A to A′. The formulation is shown in Equation (1). The light is then again projected by the reflecting surface onto the surface of the cover glass, as shown in Figure 1c, and the relationship is formulated as in Equation (2). For example, if the width of A is 1 mm and the angle of the reflective surface is 38 degrees, the length of A″ is about 4.13 mm. This means that we can inject a narrow light width (A) and get a larger light width (A″) after reflection. Based on this concept we can obtain an effective high screen ratio design. After the light is reflected by the reflecting surface, it is transmitted into the cover glass with total internal reflection (TIR). When the finger is pressed on the surface of the cover glass, the placement of the ridges will destroy the total internal reflection due to changes in the refractive index. This principle is usually called frustrated total internal reflection (FTIR). In order to allow the light to have total internal reflection in the cover glass and be partially destroyed when a fingerprint is placed on the cover glass, the angle of the reflective surface must satisfy the condition in Equation (3), where $n_{skin}$, $n_{cover\ glass}$ and $n_{air}$ are the refractive indices of fingerprint skin, cover glass and air, set to 1.5, 1.517 and 1 respectively. Thus, we get the design value of the angle of the reflection surface between 20.6 degrees and 40.7 degrees. The remaining total internally reflected light will continue to be reflected to the end of the cover glass where there is a reflective surface with the same angle, which will change the width of A′ back to the width of A. After this, a third reflective surface will be placed under the cover glass; the function of this reflecting surface is to enlarge the beam to the width of A′ and project it onto the Complementary Metal Oxide Semiconductor (CMOS) sensor. It is through the inclusion of this third reflective surface that we can obtain fingerprints on the CMOS sensor with the same area as on the cover glass. Through this design, we can create a fingerprint recognition device that can be used in opaque displays while meeting the requirement of high screen ratios.

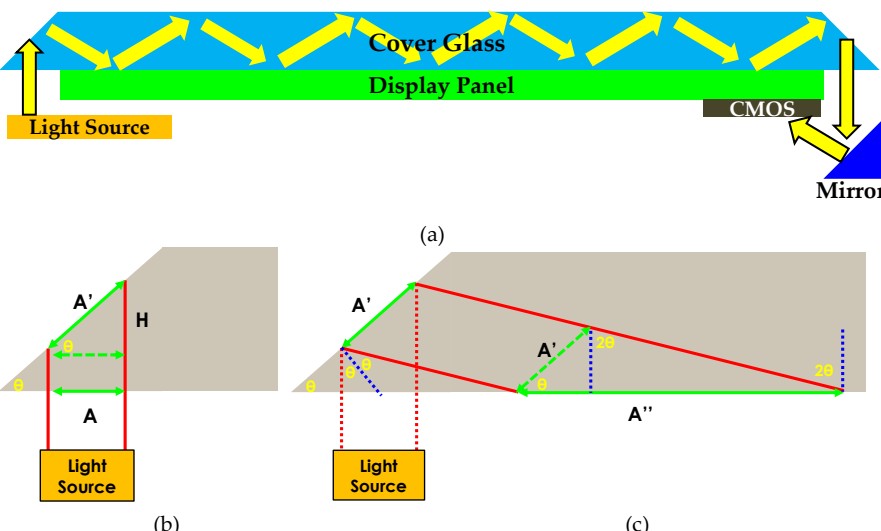

**Figure 1.** (**a**) The design concept which combines with the display panel. (**b**) Light area enlarges from A to A′ due to θ angle. (**c**) Light area enlarges from A′ to A″ due to θ angle.

$$A' = \frac{A}{\cos(\theta)} \tag{1}$$

$$A'' = A + A' \cdot \sin(\theta) \cdot \tan(2\theta) \tag{2}$$

$$n_{skin} > n_{cover\ glass} \cdot \sin 2\theta_{cover\ glass} > n_{air} \tag{3}$$

## 3. Simulation

The CMOS specifications selected for use in this paper are shown in Table 1. The shortest side of the CMOS is 4 mm, so A″ must be designed at least larger than this value. From Equation (2), the width of A″ is related to the angle of the reflective surface. This relationship is illustrated in Figure 2. The angle of reflective surface needs to be at least 37.8 degrees from the derivation of Figure 2. For instance, A 6.5-inch display with a resolution of 1920 × 1080 has a length of 143.89 cm on the long side. Therefore, the length of the cover glass must be greater than this value (143.89 cm).

**Table 1.** Specifications of Complementary Metal Oxide Semiconductor (CMOS).

| Item | Specification |
| --- | --- |
| Sensing Area | 4 mm × 6 mm |
| Pixel Size | 50 μm |
| Resolution | 80 × 120 |

The Relationship between A" width and angle

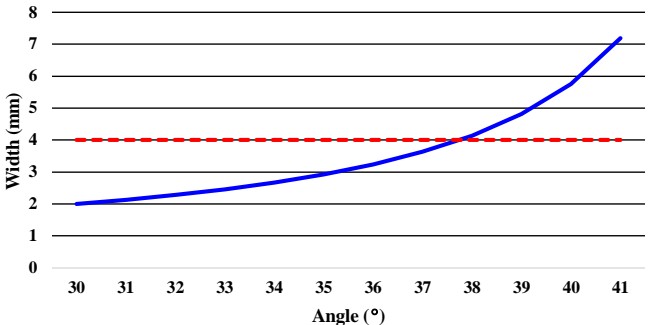

**Figure 2.** The blue line shows the enlarged relationship between A″ and the angle of the reflecting surface. The red line shows the limitation of CMOS.

Considering the above conditions, we chose a suitable design specification, shown in Table 2.

**Table 2.** Geometric specifications of cover glass.

| CG Angle | CG Thickness | CG Length | A | A′ | A″ |
| --- | --- | --- | --- | --- | --- |
| 39° | 1 mm | 147 mm | 1 mm | 1.29 mm | 4.81 mm |

Applying the design value from Table 2, the transmission of light meeting the TIR condition in the cover glass is shown in Figure 3. The light was reflected 31 times from the top to the end of the cover glass. Since the light does not meet the TIR conditions on the edge reflective surface, a reflective coating is needed. The screen ratio is an important value in the appearance standard of mobile phones. At present, the screen ratio of high-end mobile phones on the market needs to be more than 90%. The definition of screen ratio is shown in Equation (4). Using the cover glass length value given in

Table 2, for a 6.5-inch display, the screen ratio value obtained with our design can satisfy the demand for a ratio of more than 90%.

$$\text{Screen Ratio} = \frac{Area\ of\ Display}{Area\ of\ Cover\ Glass} \tag{4}$$

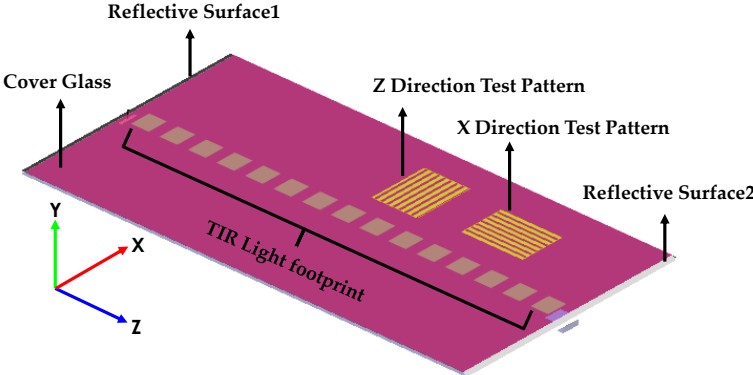

**Figure 3.** shows that light is transmitted through the cover glass by total internal reflection.

The MTF is usually an indicator for judging the quality of the camera image. In this paper, the contrast is used instead. Because the fingerprint algorithm performs a series of image processing steps on the image in advance, only the fringe information of the ridges or valleys is left. However, the quality of these image processing will have a strong relationship with the gray difference between the ridges and valleys of the original pattern. Therefore, contrast was chosen as the main quality indicator in this paper. The formulated as in Equation (5).

$$\text{Contrast} = \frac{I_{max} - I_{min}}{I_{max} + I_{min}} \tag{5}$$

The width of the ridges and valleys of fingerprints is usually between 0.1 and 0.4 mm, and there are many types of ridges and valleys of the fingerprint—the core, delta, fork, etc. Therefore, we divide the direction of fingerprint fringe in x direction and z direction, as shown in Figure 3. The periods between fingerprint ridges and valleys are assumed to be 0.1, 0.2, 0.3 and 0.4 mm, respectively, and the divergence angle of the light source is ±0.5 degrees. Therefore, the analysis of fingerprint width and image contrast is shown in Figure 4.

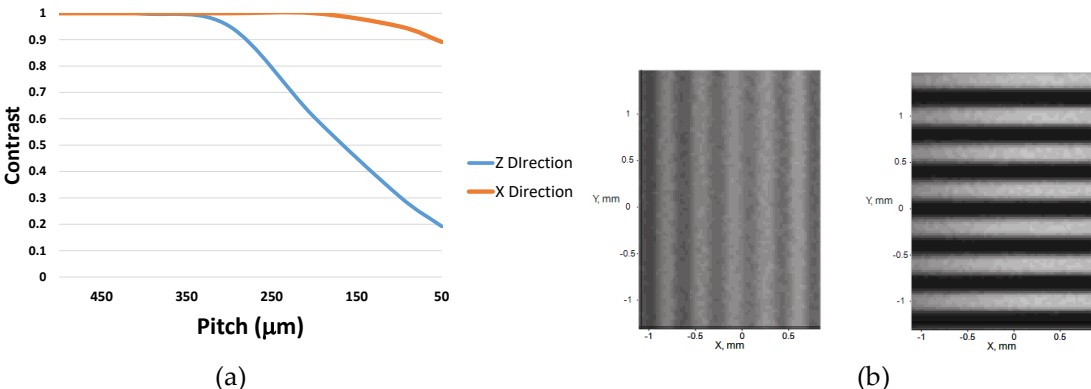

(a)　　　　　　　　　　　　　　　　　　(b)

**Figure 4.** (**a**) Analysis chart of contrast when the light source divergence angle is 0.5 degrees and the fingerprint periods in the x direction and the z direction are changed. (**b**) Energy distribution of the fingerprint ridge and valley on the CMOS sensor.

As can be seen in Figure 4, the contrast in the X direction is higher than the Z direction in the same fingerprint width, because the X component of the light source angle cannot be incident on the CMOS sensor, as the transmission distance is longer, as shown in Figure 5a. For example, the divergence angle of the light source is 1 degree, but the angle that can actually be received by the CMOS sensor is only 0.5 degrees. However, the X component of the light source angle will be incident on the CMOS sensor regardless of the transmitted distance, as shown in Figure 5b. Figure 5b also shows that light of different angles is reflected into the cover glass at different frequencies. Therefore, for the same fingerprint pressing area, the light of different angles will obtain the same fingerprint signal at the same time. This will cause a rapid decrease in contrast on the CMOS sensor, as shown in Figure 6. With this design, only the optical magnification in the Z direction is corrected; the magnification in the X direction is not modified. In order to avoid image distortion, in future, the Y direction light can be designed to be as parallel as possible, or algorithms can be used to correct the image distortion.

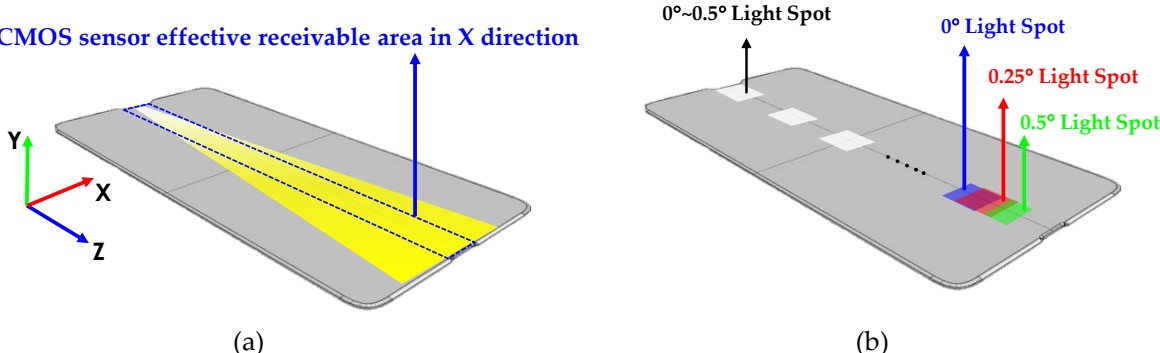

(a)　　　　　　　　　　　　　　　　　　　　　　　　　　(b)

**Figure 5.** (**a**) Schematic diagram of light transmission area when propagating in the z direction. The blue box shows the range that the CMOS sensor can receive. (**b**) When the light is transmitted in the Z direction, the light spots at each angle gradually becomes separate from the transmission distance.

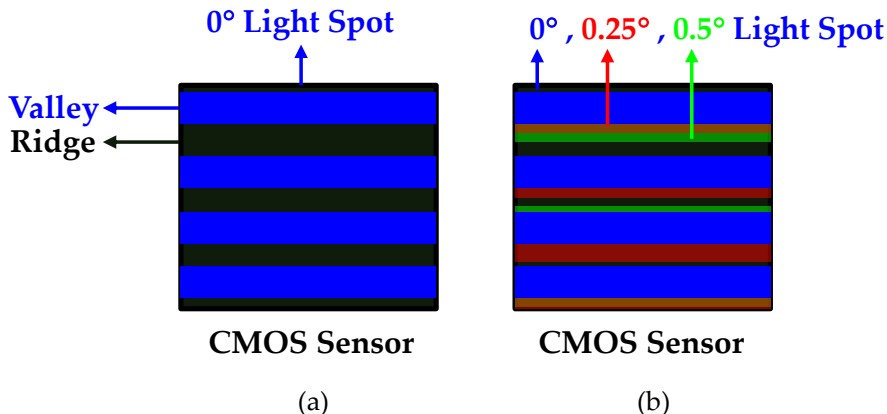

(a)　　　　　　　　　　　　　　　(b)

**Figure 6.** (**a**) Schematic diagram of the energy distribution on the CMOS sensor after absorption of 0° light from the fingerprint ridge signal. (**b**) Schematic diagram of the energy on the CMOS sensor after absorption of the 0, 0.25, and 0.5 degree light from the fingerprint ridge signal. Because the reflection frequency of each angle is different, the gray level of the ridge changes, resulting in a decrease in contrast.

## 4. Fabrication

According to the design principles and simulation results, we made a piece of cover glass with the specifications shown in Table 2. The material of the cover glass here is Corning GG3 glass. First, we use computer numerical control (CNC) processing technology to complete the glass shape processing (such as the upper and lower notch parts and lead angle around it), and then polish the optical surface of the notch part to make the surface a sufficiently smooth optical surface. We have

optimized the appearance of the reflective surface 1 and the reflective surface 2. Using the notch design method, the screen utilization can be maximized, as shown in Figure 7. The silver part is coated with aluminum metal, and a lead angle is added around the cover glass to avoid cracking during process.

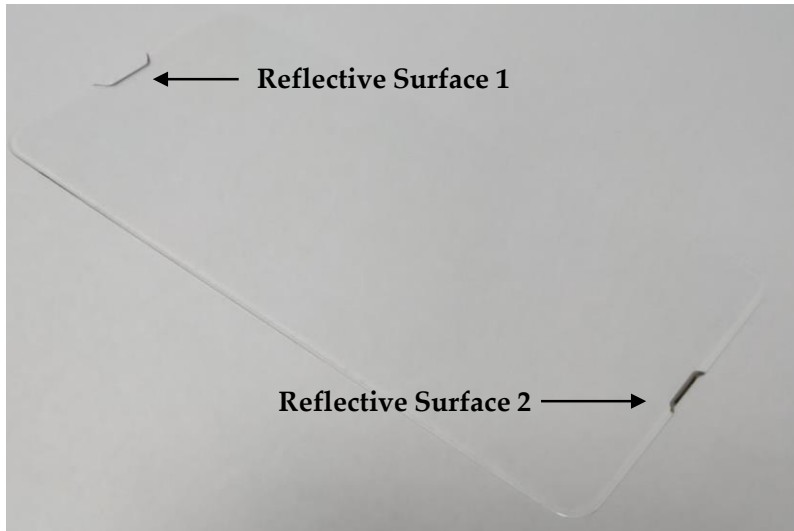

**Figure 7.** Cover glass processed according to Table 2 specifications, with lead angles added around the sides to avoid cracking.

Figure 8a is a cross-sectional view of reflective surface 1. We used Panasonic's Ultrahigh Accurate 3-D Profilometer (UA3P) to measure the surface condition and angle of the inclined surface. The result is shown in Figure 8b. The angle of the reflecting surface 1 is 39.09 degrees. From the cross-sectional view of Figure 8a, we can see the design of the lead angle on the upper and lower sides of the bevel. This lead angle design is necessary; otherwise the bevel will be easily broken during processing. The cross-sectional view of the reflecting surface 2 is shown in Figure 9a. In the design, the angles of the two reflecting surfaces are the same. The actual measured angle using UA3P is 39.01 degrees. The UA3P cannot measure the lead angle extending inward, so the bottom part of the measurement chart and the actual section view are different.

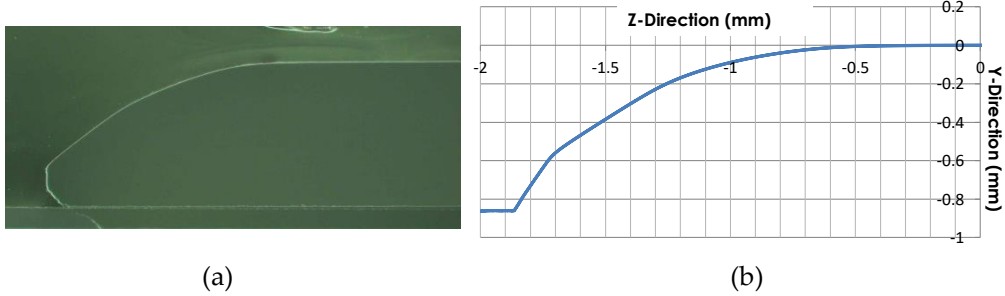

(a)　　　　　　　　　　　　　　　　　　　　　　　　　　　(b)

**Figure 8.** (**a**) Cross-sectional view of reflection surface 1 of the cover glass. (**b**) Using UA3P measurement results, the angle of reflection surface 1 is 39.09 degrees.

An HeNe laser is used to enter the cover glass after expanding and collimating. The characteristics of total reflection of light in the glass are shown in Figure 10a. From the observations, the beam does not diffuse significantly in the X direction due to the HeNe laser being collimated, and both angle components of the light have collimation characteristics. Figure 10b shows the actual pressure of the finger. Since the FTIR concept is applied in the design, the finger pressure contrast between ridges and valleys is clear and easy to recognize. In Figure 10b, the black parts are ridges and the white parts are valleys. In addition, we use infrared LED as the light source. The smaller size of the LED is easier

to integrate into the phone than a laser, but the divergence angle is larger, so the contrast between ridges and valleys is poor, as shown in Figure 10c.

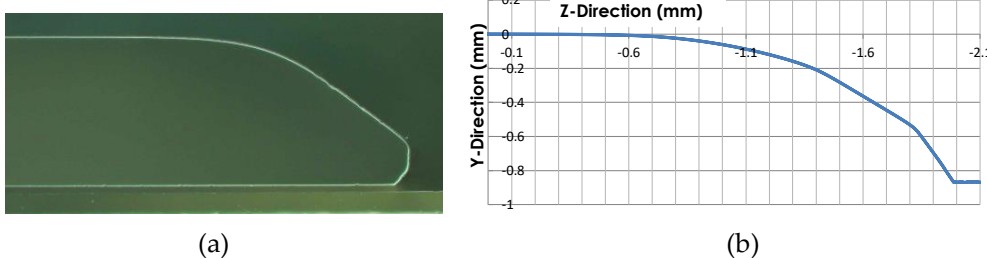

(a)    (b)

**Figure 9.** (**a**) Cross-sectional view of reflection surface 2 of the cover glass. (**b**) Using UA3P measurement results, the angle of reflection surface 2 is 39.09 degrees.

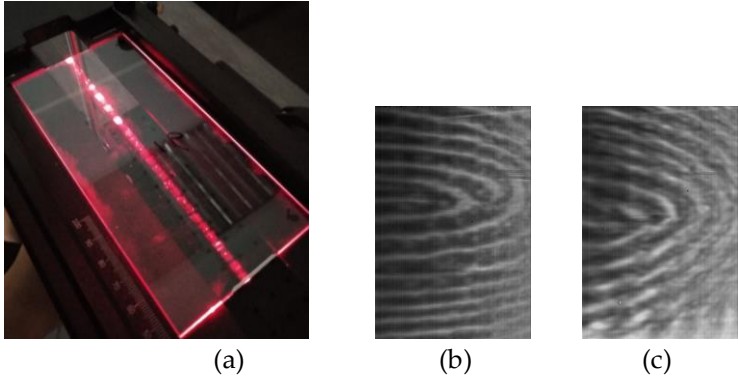

(a)    (b)    (c)

**Figure 10.** (**a**) HeNe laser performs light transmission with total internal reflection in the cover glass. (**b**) The actual fingerprint pressing pattern, where the black parts are ridges and the white parts are valleys. (**c**) The actual fingerprint image formed by infrared LED light source.

## 5. Conclusions

We designed and fabricated an OFR system based on the characteristics of the cover glass on the opaque display panel. Through application the waveguide and FTIR methods, the problem of the opacity of the display panel can be avoided, by covering both ends of the cover glass. In addition, a reflective surface is designed and coated at each end of the cover glass, and this reflective surface can enlarge and reduce the signal. With this design, we can achieve a screen ratio greater than 90%. Multiple reflections of light in the cover glass can form multiple identifying places, allowing users to have more different fingerprint recognition locations using the same CMOS sensor. Finally, our research has developed a new fingerprint identification device method that can meet the optical identification fingerprint requirements of opaque panel mobile phones.

**Author Contributions:** J.-S.L. and W.-S.S. conceived and designed the experiments; J.-S.L. performed the experiments; J.-S.L. analyzed the data; W.-S.S. contributed reagents/materials/analysis tools; J.-S.L. wrote the paper. All authors have read and agreed to the published version of the manuscript.

**Funding:** This research was funded by Ministry of Science and Technology of Taiwan grant number MOST 108-2221-E-008-090 and the APC was funded by MOST 108-2221-E-008-090.

**Acknowledgments:** This study was sponsored by the Ministry of Science and Technology of Taiwan, under project number MOST 108-2221-E-008-090.

**Conflicts of Interest:** The authors declare no conflict of interest.

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
