# Peer review of "A Hidden Fingerprint Device on an Opaque Display Panel"

_applsci, doi:10.3390/app10062188_

Round 1

Reviewer 1 Report

The manuscript "Hidden Fingerprint Device on Opaque Display Panel" proposed a method using the concept of frustrated total internal reflection to project an image of a fingerprint onto a CMOS camera for mobile phone displays. They designed and demonstrated that the addition of a coverglass with reflective surfaces can be used to enlarge and reduce the image.

The work is interesting and potentially can be implemented in mobile phone displays for fingerprint recognition due to its simplicity. The design and implementation are convincing. I summarise my comments and suggestions below:

1) The authors should cite work by Jefferson Han, "Low Cost Multi-Touch Sensing through Frustrated Total Internal Reflection" who published a similar concept back in 2005, although not specific for mobile phone displays. The implementation of the authors of course are different and the authors could further expound on this in their Introduction.

2) Additionally, it would also be interesting to the readers to know what are other optical methods that have been implemented to image fingerprints such as optical coherence tomography etc. Again in mobile phone displays, this would be difficult to implement due to bulky optics but it would be a good introduction on the methods that have been used. Also, could the authors mention what is the current method used for fingerprint reading in mobile phones and how would an optical method compare to this?

3) The authors did not mention how they manufacture the coverglass with the reflective surfaces. Could the authors elaborate on this in their methods?

4) Is contrast the only measure one can use to quantify the images obtained at the camera? Could the authors also use fidelity or other measures to be able to  quantify the performance of their design?

5) Have the authors tested other light sources - not only a collimated laser but for example LED for their approach? LEDs might be more practical to use due to their compactness and therefore it would be a good step to test if this also provides a good image of the fingerprint.

Some other minor comments:

The word "Laser" should not be all capitalized

Author Response

Thanks for your suggestions, all the replies are attached in the attachment.

Reviewer 2 Report

Please make the readings in Fig 8b & 9b readable.

The article is good. The device is basically operable. I did not find direct plagiatum. However I do not understand why the authors did not patent it?

Author Response

(The authors gave the same response as above.)
